

# Radiosensitization effect of hsa-miR-138-2-3p on human laryngeal cancer stem cells

Ying Zhu[1,*], Li-Yun Shi[2,*], Yan-Min Lei[3], Yan-Hong Bao[3], Zhao-Yang Li[4], Fei Ding[4], Gui-Ting Zhu[4], Qing-Qing Wang[5] and Chang-Xin Huang[4]

[1] First Affiliated Hospital, Zhejiang Chinese Medical University, Hangzhou, China
[2] Department of Immunology, School of Medical and Life Science, Nanjing University of Chinese Medicine, Nanjing, China
[3] Second Clinical Medical College, Zhejiang Chinese Medical University, Hangzhou, China
[4] Department of Oncology, Affiliated Hospital with Hangzhou Normal University School of Medicine, Hangzhou, China
[5] Institute of Immunology, Zhejiang University, Hangzhou, China
[*] These authors contributed equally to this work.

Corresponding author
Chang-Xin Huang,
hcx588@hotmail.com,
hcx588@tom.com

## ABSTRACT

**Background**. Treatments that target cancer stem cells play an important role in the controlling and eliminating of tumor initiation as well as in development, progression, and chemotherapy/radiotherapy resistance. In our previous study, we cultured and harvested human laryngeal cancer stem cells (CSCs) and applied microRNA biochips to screen differentially expressed miRNAs that were related to radiation tolerance in irradiated human laryngeal CSCs. According to the predicted genes and pathways of differential miRNAs target, down-regulated expression of hsa-miR-138-2-3p under radiation was thought to play a key role in enhancing the radio-sensitivity in human laryngeal squamous cancer stem cells.

**Method**. To investigate the radiational enhancement of hsa-miR-138-2-3p, we transfected hsa-miR-138-2-3p mimics that were synthesized based on the sequences of hsa-miR-138-2-3p *in vitro* into human laryngeal CSCs (Hep-2, M2e, and TU212 cell lines) to make hsa-miR-138-2-3p overexpressed, and the tumorous specialities of CSCs, like cell proliferation, invasion, apoptosis, cell cycle arrest, and DNA damage were evaluated by CCK-8 assay, clone formation assay, invasion assay, flow cytometry, and comet assay. Furthermore, we explored the signal transduction pathways that regulated the cancer stem cell initiation, development, invasion, apoptosis and cell cycle arrest, which were controlled by hsa-miR-138-2-3p.

**Result**. Overexpressed hsa-miR-138-2-3p played a key role in many anti-cancer biological processes in human laryngeal CSCs: (1) it decreased laryngeal CSCs proliferation and invasion in response to radiotherapy; (2) it increased the proportion of early and late apoptosis in laryngeal CSCs after radiation, raised G1 phase arrest in laryngeal CSCs after radiation, and decreased the proportion of S stage cells of cell cycle that were related to radio-resistance in laryngeal CSCs; (3) it down-regulated the expression of $\beta$-catenin in Wnt signal pathway that was related to the tolerance of laryngeal CSCs to radiotherapy; (4) it down-regulated the expression of YAP1 in Hippo signal pathway that regulated cell proliferation, invasion and apoptosis; (5) it up-regulated the expression of p38 and JNK1 in MAPK signal pathway that was concerned to radio-sensitivity.

**Conclusion**. In the present study, it was found that hsa-miR-138-2-3p regulated the Wnt/$\beta$-catenin pathways, the Hippo/YAP1 pathways, and the MAPK/p38/JNK1 pathways that were involved in cell proliferation, invasion, apoptosis, cell cycle arrest, radio-resistance and radio-sensitivity in laryngeal CSCs. These results will be useful for a better understanding of the cell biology of hsa-miR-138-2-3p in laryngeal CSCs, and for serving hsa-miR-138-2-3p as a promising biomarker and as a target for diagnosis and for novel anti-cancer therapies for laryngeal cancers.

# INTRODUCTION

Many studies have shown that the biological characteristics of tumors, including radiation tolerance, are determined by tumor stem cells. *Rycaj & Tang (2014)* showed that breast cancer and malignant glioma were tolerant to radiotherapy mainly due to the existence of cancer stem cells, which had powerful DNA repair capacity and defensive ability against reactive oxygen species and strong self-renewal capacity. *Peitzsch et al. (2013)* indicated that the plasticity of cancer stem cells would be beneficial under the stress of various factors, and cancer stem cells (CSCs) were involved in reducing radio-sensitivity by changing their structures to generate radiation-induced resistance. *Colak & Medema (2014)* found that the nature of radiation resistance of tumor cells were cancer stem cells that prevented cancer cell apoptosis and enhanced the efficiency of DNA repair. Therapies targeting CSCs are necessary for controlling and eliminating tumor cell growing, invasion, progression and radio-resistance. *Kobayashi et al. (2014b)* found that the removal of Brachyury that was related to cell proliferation, migration, invasion, radiation and chemotherapy resistance of Adenoid Cystic carcinoma CSCs could improve tumor stem cell radiation sensitivity. *Wu et al. (2012a)* developed a model of CDK1 knock-out brain malignant glioma and found that CDK1 contributed to enhance radiosensitivity of CSCs by inducing cell apoptosis. Cell cycle checkpoint kinases (CDK1 and CDK2) play key roles in DNA damage response in radiation and chemotherapy.

MicroRNA (miRNA), a non-coding oligonucleotide single chain of ∼20nt in length, was recently considered as the most important gene regulation factor in cancer cells, and may improve the radiosensitivity of tumor cells (*Zhao et al., 2012*). *Jiang et al. (2015)* found that increased expression of the miR-17-92 cluster could enhance the resistance to radiation of lymphoma. *Wu et al. (2012b)* demonstrated that MiR-148b could enhance radiosensitivity of non-Hodgkin's lymphoma cells by promoting apoptosis. However, the role of hsa-miR-138-2-3p in laryngeal CSCs has not been not reported before.

In our previous study (*Huang et al., 2013*), we used microRNA biochips to compare and screen the differential expression microRNAs in laryngeal CSCs in response to radiation stress. Based on the predicted genes and pathways of miRNA target, the expression profile of hsa-miR-138-2-3p that was down-regulated significantly after radiation was thought to

play an important role in regulation of radio-sensitivity in laryngeal squamous CSCs. In the present study, we synthesized hsa-miR-138-2-3p *in vitro* and transfected it into three types of laryngeal CSCs (Hep-2, M2e, TU212) to make hsa-miR-138-2-3p overexpressed, and evaluated the tumorous specialities of CSCs, such as cell proliferation, invasion, apoptosis, cell cycle arrest, and DNA damage. Furthermore, we explored the signal transduction pathways that were involved in cell initiation, development, invasion, apoptosis and cell cycle arrest, which were regulated by hsa-miR-138-2-3p. These results will be useful for a better understanding of cell biology of hsa-miR-138-2-3p in laryngeal CSCs, and serve hsa-miR-138-2-3p as a promising biomarker and target for diagnosis and for novel anti-cancer therapies for laryngeal cancers.

## MATERIALS AND METHODS

### Laryngeal cancer sphere culture

Three human laryngeal squamous cancer cell lines, Hep-2, TU212 and M2e, were obtained from the American Type Culture Collection (ATCC, Manassas, VA, USA). Serum supplement medium (SSM) contained 90% RPMI-1640 (Gibco, Waltham, MA, USA) and 10% fetal bovine serum (Gibco). Serum free medium (SFM) contained DMEM/F12 (Gibco); and 4 mg/ml heparin; 10 ng/ml basic fibroblast growth factor (bFGF; Peprotech, Rocky Hill, NJ, USA), 20 ng/ml epidermal growth factor (EGF; Peprotech, Rocky Hill, NJ, USA); 25 mg/ml insulin; and 2ml 50X B27 supplement (Gibco). Cells in exponential growth phase were washed with PBS (Gibco) and digested with 0.25 trypsin/0.02% ethylenediaminetetraacetic acid (EDTA; Gibco), followed by resuspension in SFM at a concentration of 5X10E5 cells/ml. The medium was changed every 5 days in half amount. Each cell line was regularly observed to confirm its morphology and absence of mycoplasma contamination.

### Sorting of laryngeal CSCs based on cell surface marker expression

The laryngeal cancer sphere of Hep-2, M2e and TU212, was digested, a single-cell suspension was prepared and the cell number was counted before labeling. Cells were collected by centrifuge at 1000 rpm for 5 min and the cell pellets were resuspended in 90ul of PBS buffer per 10E7 total cells. 10ul of anti-human-CD133-FITC (AC-133-FITC, mouse IgG1, Miltenyi, Germany) were added. The samples were mixed well and incubated in the dark for 30 min at 4 °C refrigerator. The analysis was performed with FACS caliber (BD, Franklin Lakes, NJ, USA), and CD133 positive expression cells were investigated as laryngeal CSCs.

### Hsa-miR-138-2-3p targets prediction

In our earlier research (*Huang et al., 2013*), laryngeal CSCs were harvested and accepted to radiation stress. We applied microRNA biochips to identify and screen differential expression miRNAs, and more than 2-fold up-regulation/down-regulation expression were considered as differential expressions. Meaningful miRNAs were selected by targeted genes from Targetscan Human 6.2 (http://www.targetscan.org; *Lewis, Burge & Bartel, 2005*) and miRanda (http://www.microrna.org/microrna/home.do; *Betel et al., 2008*). The sequences

of miRNAs were inquired from miRBase (http://www.miRbase.org; *Kozomara & Griffiths-Jones, 2014*). To understand the targeted biological process, we applied starBase v2.0 (http://starbase.sysu.edu.cn/index.php; *Li et al., 2014*) to analyze signal transduction pathways that were regulated by microRNAs from pathway databases (e.g., GO, KEGG, BIOCARTA). Hsa-miR-138-2-3p mimics, nonsense oligonucleotides, and negative control FAM oligonucleotides with fluorescence were synthesized *in vitro* (Invitrogen, Shanghai, China).

## Transient cell transfection

Laryngeal CSCs ($2 \times 10E5$ cells/ well) were plated in 12-well culture plates, and were transfected equal volume with gradient concentrations of hsa-miR-138-2-3p mimics (conc: 50 nM, 100 nM, 150 nM). Nonsense oligonucleotides (conc: 100 nM), negative control FAM oligonucleotides (conc: 100 nM), and PBS buffer with the same volume as hsa-miR-138-2-3p were transfected into laryngeal CSCs. The hsa-miR-138-2-3p teams with gradient concentration were considered as experimental team and were named as 50nM-TR, 100nM-TR, 150nM-TR, respectively. Nonsense oligonucleotides team, negative control FAM oligonucleotides team, and PBS buffer team were considered as control teams, and were named as 100nMN-CR, FAM-CR, and PBS-CR. All the teams were added in Entranster$^{TM}$-R transfection reagent (Engreen Biosystem, Beijing, China) and mixed sufficiently, according to the manufacturer's instructions. All teams were with the final concentrations of 50 nM per well. After mixing, all 12-well culture plates were incubated for 6 h at 4 °C refrigerator. The transfection efficiency of hsa-miR-138-2-3p mimics and nonsense oligonucleotides were evaluated by the positive expression of negative control FAM oligonucleotides by flow cyometry.

## Irradiation

Laryngeal CSCs were irradiated by a linear accelerator with a 6-MV X ray. Culture plates were placed under a 15 mm tissue equivalent filler. The distance between filler and radiation source was 100 mm. Experimental teams and control teams were irradiated continuously at total does of 2Gy each day, for 2 days. 100nM-T, 100nMN-C, PBS-C teams were treated without radiation and were placed out the range of radiation at the same time. All the three teams were served as control teams.

## Proliferation assays

All the experimental and control teams were respectively resuspended in 0.1 ml SFM at a density of 5000 cells/well in 96-well microwell culture plates. Cell proliferation were analyzed at 0 h, 24 h, 48 h and 72 h after the second radiation by CCK-8 assay (Cell Counting Kit-8; Engreen Biosystem, Beijing, China) (*Huang et al., 2012b*). Viable cells were quantified by measuring absorbance at 450 nm absorption spectra in a microplate reader, and were named as "A450 value".

## Apoptosis and cell cycle assay

For apoptosis assay, 100nM-TR and 100nMN-CR of Hep-2, M2e, and TU212 CSCs were resuspended in PBS buffer by the amount of cells 5000/ml. 195 ul cell suspension were mixed well with 5 ul Annexin V-FITC and incubated at room temperature for 10 min.

Cells were washed with PBS and resuspended in 190 ul deliquated binding buffer, then 10 ul 20 ug/ml PI were added. Identified by flow cytometry, cells were divided into four sections: Q1: Annexin V-FITC - PI +, was representative of mechanical error; Q2: Annexin V-FITC+ PI+, was representative of late apoptosis or necrosis cells; Q3: Annexin V-FITC - PI -, was representative of living cells; Q4: Annexin V-FITC+ PI-, was representative of early apoptosis cells.

For cell cycle analysis, 100nM-TR and 100nMN-CR of Hep-2, M2e, and TU212 CSCs were washed with cold PBS for three times, and then the cells were fixed in 70% ethanol at −20 °C for 12 h. Following the fixation, cells were washed with cold PBS and stained with 500 ul PI at 37 °C in dark for 30min. Analyses were performed on flow cytometry.

## Invasion assays

For Transwell migration assay, 12-well Transwell chambers containing 8 um pores were coated with 700 ul Matrigel (BD, Franklin Lakes, NJ, USA) at −20 °C. According to the manufacturer's instructions (*Wang et al., 2014b*), 100nM-TR and 100nMN-CR of Hep-2, M2e, and TU212 CSCs were collected and resuspended in 1 ml DMEM/F12 at a density of 2X10E5 cells/ ml. A total of 1 ml cell suspension of six teams were added to the upper Transwell chamber, and 600 ul SFM were added to the lower chamber. After 24 h of cell incubation, cells that were cultured previously above the upper chambers, migrated to the other side of the upper chambers. The reverse side of the upper chambers were fixed with 4% paraformaldehyde and stained with crystal violet. The numbers of migrated cells on the reverse side were counted at least five random microscopic fields by a light microscope at a magnification of 200X (Olympus, Tokyo, Japan).

## Clone formation assay

100nM-TR, 100nMN-CR, and non-transfection teams of Hep-2, M2e, and TU212 CSCs that proliferated at exponential growth phase, were digested and cultured at the density of 10E6/dish in 10 cm petri dishes. After 24 h of culture, cells were treated with 0, 2, 4, 6, 8 Gy X-ray irradiation, respectively, and then were incubated for additional 14 days. After incubation, cells were washed with PBS buffer for three times, and fixed with paraformaldehyde for 20 min and air dried overnight. Cells were then washed with water and the colonies (>50 cells) were counted for each culture dish by a light microscope (Olympus, Tokyo, Japan).

## Comet assay

100nM-TR and 100nMN-CR of Hep-2, M2e, and TU212 CSCs at equal volume of cell suspension (4X10E5 cells) were mixed with 0.5% (w/v) low melting agarose (LMA) in 0.01 M PBS buffer, respectively. The mixture were pipetted on the frosted slides with pre-coating of normal melting agarose 1% (w/v). After the agarose solidified, another 100 μl of 0.5% (w/v) LMA were pipetted on the slides and immersed in lysis buffer (2.5 M NaCl, 100 mM EDTA, 10 mM Tris–HCl buffer, 0.1% SDS and 1% Triton X-100 and 10% DMSO; pH 10.0) for 120 min in dark at 4 °C to lyse the cellular and nuclear membranes. The slides were rinsed with unwinding buffer and transferred into an electrophoresis tank containing unwinding buffer (3 M NaOH, 10 mM EDTA; pH 13.0) for denaturing the DNA followed

**Table 1  Target genes of hsa-miR-138-2-3p.**

| Target gene | Representative transcript | Gene name |
|---|---|---|
| MAP3K11 | NM_002419 | mitogen-activated protein kinase kinase kinase 11 |
| ARHGEF3 | NM_001128615 | Rho guanine nucleotide exchange factor (GEF) 3 |
| HIF1AN | NM_017902 | hypoxia inducible factor 1, alpha subunit inhibitor |
| CASP3 | NM_004346 | caspase 3, apoptosis-related cysteine peptidase |
| ACVR2B | NM_001106 | activin A receptor, type IIB |

by electrophoresis for 30 min with an electric current of 25 V. The slides were washed twice with neutralizing buffer (0.4 M Tris–HCl; pH 7.5) for 10 min and ethanol treatment was done another 5 min. Ethidium bromide (20 mg/ml) 40 µl were used to stain the slides and DNA damage visualized using fluorescence microscope (Olympus, Tokyo, Japan). Appearance of 'comet' with fragmented DNA (tail) being separated from undamaged nuclear DNA (head) was seen in damaged cells and measurements were made by Comet Assay IV software to determine the tail movement (%). The results were expressed as percent tail movement.

## Western blotting

The total proteins of 100nM-TR, 100nMN-CR, and non-transfection teams of Hep-2, M2e, and TU212 CSCs were separated by 10% sodium dodecyl sulfate polyacrylamide gel electrophoresis (SDS-PAGE) and transferred to membranes. The membranes were blocked in 5% BSA, diluted in 1X  TBS-Tween for 1 h and then incubated overnight with anti-human $\beta$-catenin/YAP1/p38/JNK1 antibody (Abcam, Cambridge, UK) according to the manufacturer's instructions. Primary antibody binding was detected with secondary IgG-HRP antibodies goat anti-rabbit (Abcam, Cambridge, UK). Actin was used as control. Images were captured on MicroChemi 4.2 (Eastwin, Shenzen, China).

## Statistical analysis

Data are shown as mean $\pm$ standard deviation ($\bar{x} \pm s$). One-way and two-way ANOVA analyses were applied to compare the sample means of test groups and control groups. The Fisher's exact test was applied to compare the sample rates of test groups and control groups. All statistical analyses were performed using SPSS 17.0 software (SPSS Inc., Chicago, IL, USA). $P < 0.05$ was considered statistically significant.

## RESULTS

### Hsa-miR-138-2-3p was selected to transfected into the laryngeal CSCs

Based on our previous study (*Huang et al., 2013*), we applied Targetscan and Miranda to investigate target genes of hsa-miR-138-2-3p, such as MAP3K11, CASP3 and HIF1AN, Which were involved in cell apoptosis, radio-sensitivity, and cell cycle arrest (Table 1).

The sequences of hsa-miR-138-2-3p were inquired from miRBase, and hsa-miR-138-2-3p mimics, nonsense oligonucleotides, and negative control FAM oligonucleotides with fluorescence were synthesized *in vitro* (Invitrogen, Shanghai, China). The oligonucleotide sequences were listed in Table 2.

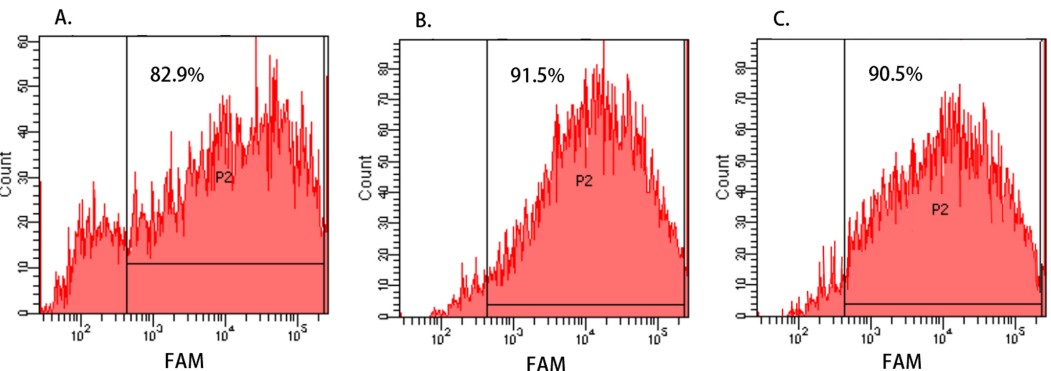

**Figure 1  Cell transfection efficiency evaluated by Flow Cytometry.** (A–C) show the transfection efficiency of FAM-CR teams of Hep-2, M2e, and TU212 cell line, respectively, and the percentages of FAM-positive cells (represented by "P2") were identified by flow cytometry. The transfection efficiency of Hep-2, M2e, and TU212 cell lines were 82.9%, 91.5% and 90.5%, respectively, and it was indicated that hsa-miR-138-2-3p and nonsense oligonucleotides were successfully transfected into the laryngeal CSCs with high efficiency.

**Table 2  Sequence of synthesized oligonucleotides *in vitro*.**

| Oligonucleotides | Sequence |
| --- | --- |
| hsa-miR-138-2-3p mimics | 5′- GCUAUUUCACGACACCAGGGUU - 3′ |
| nonsense oligonucleotides | 5′-AAGGCAAGCUGACCCUGAAGU-3′<br>3′-UUCAGGGUCAGCUUGCCUUUU- 5′ |
| Fluorescein FAM tag oligonucleotides | 5′-UUCUCCGAACGUGUCACGUTT-3′<br>3′- ACGUGACACGUUCGGAGAATT- 5′ |

To evaluate the transfection efficiency of hsa-miR-138-2-3p and nonsense oligonucleotides, the FAM-CR teams of Hep-2, M2e, and TU212 cell lines that were transfected with FAM-labeled oligonucleotide were digested and resuspended in PBS buffer. The percentages of FAM-positive cells were identified by flow cytometry. As shown in Fig. 1, the transfection efficiency of Hep-2, M2e, and TU212 cell lines were 82.9%, 91.5% and 90.5%, respectively, and it was indicated that hsa-miR-138-2-3p and nonsense oligonucleotides were successfully transfected into the laryngeal CSCs with high efficiency.

## Overexpressed hsa-miR-138-2-3p was involved in various biological process to enhance the radiosensitivity of laryngeal CSCs

### Overexpressed hsa-miR-138-2-3p inhibited cell proliferation after radiation

The cell proliferation rates of each team at 0, 24, 48 and 72 h after radiation were shown in Fig. 2. From Fig. 2A, we can infer that at 48 h after radiation, the cell proliferation rate of 50nM-TR, 100nM-TR, and 150nM-TR were lower than 100nMN-CR and PBS-CR. But at 0 h, 24 h and 72 h after radiation, the difference were not observed ($P > 0.05$). It was noted that at 48 h after radiation, the cell proliferation rate of 50nM-TR, 100nM-TR, and 150nM-TR were slower than 100nMN-CR and PBS-CR, the differences were statistically significant ($P < 0.001$, $P < 0.001$, $P < 0.001$), and the inhibition capacity of 100nM-TR and 150nM-TR were stronger than 50nM-TR. From Fig. 2B, it is seen that without radiation,

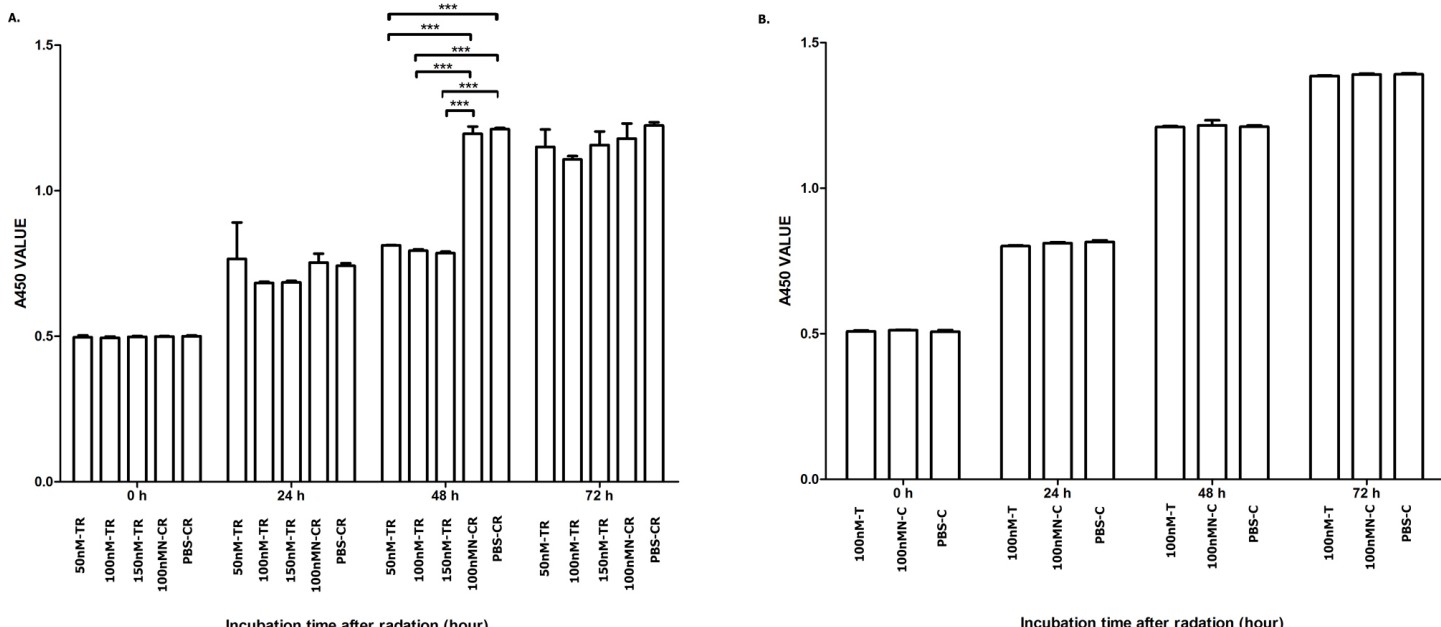

**Figure 2** **Overexpressed hsa-miR-138-2-3p inhibit cell proliferation after radiation.** (A) shows the comparison of cell proliferation among 50nM-TR, 100nM-TR, 150nM-TR, 100nMN-CR and PBS-CR of Hep-2 cell line after radiation. (B) shows the comparison of cell proliferation among 100nM-T, 100nMN-C and PBS-C of Hep-2 cell line without radiation. The vertical and horizontal axis stand for A450 (absorbance at 450 nm absorption spectra) value and time after radiation, respectively. From (A), we can infer that at 48 h after radiation, the cell proliferation rate of 50nM-TR, 100nM-TR, and 150nM-TR were lower than 100nMN-CR and PBS-CR. But at 0 h, 24 h and 72 h after radiation, the difference were not observed ($P > 0.05$). It was noted that at 48 h after radiation, the cell proliferation rate of 50nM-TR, 100nM-TR, and 150nM-TR were slower than 100nMN-CR and PBS-CR, the differences were statistically significant (*** $P < 0.001$), and the inhibition capacity of 100nM-TR and 150nM-TR were stronger than 50nM-TR. (B) shows that without radiation, the differences of cell proliferation rate of 100nM-T, 100nMN-C, and PBS-C were not statistically significant ($P > 0.05$). Date are reported as mean $\pm$ SD.

the differences of cell proliferation rate of 100nM-T, 100nMN-C, and PBS-C were not statistically significant ($P > 0.05$).

### Overexpressed hsa-miR-138-2-3p induced cell apoptosis after radiation

To investigate whether the declined cell proliferation was due to the cell apoptosis, we used flow cytometry analyses to evaluate the effect of overexpressed hsa-miR-138-2-3p on promotion of cell apoptosis. We found that the proportion of early apoptosis and late apoptosis of Hep-2, M2e, and TU212 CSCs induced by transfection of hsa-miR-138-2-3p were larger than that induced by transfection of nonsense oligonucleotides, the differences between them were statistically significant ($P < 0.001$)(Figs. 3 and 4).

### Overexpressed hsa-miR-138-2-3p induced cell cycle arrest after radiation

DNA content stained with PI were detected with flow cytometry, and cell cycle distribution of Hep-2, M2e, and TU212 cell lines were analyzed by ModFit software (*Nojiri & Joh, 2014*). As Fig. 5 shown, the growth cycle of cell is divided into five stages, known as G0 phase, in which cell is quiescent, G1 and G2 phase, in which it increases in size, S phase, in which it duplicates its DNA, and M phase, in which it undergoes mitosis and divides. As shown in Figs. 6 and 7, the percentage of Hep-2, M2e, and TU212 CSCs induced by transfection

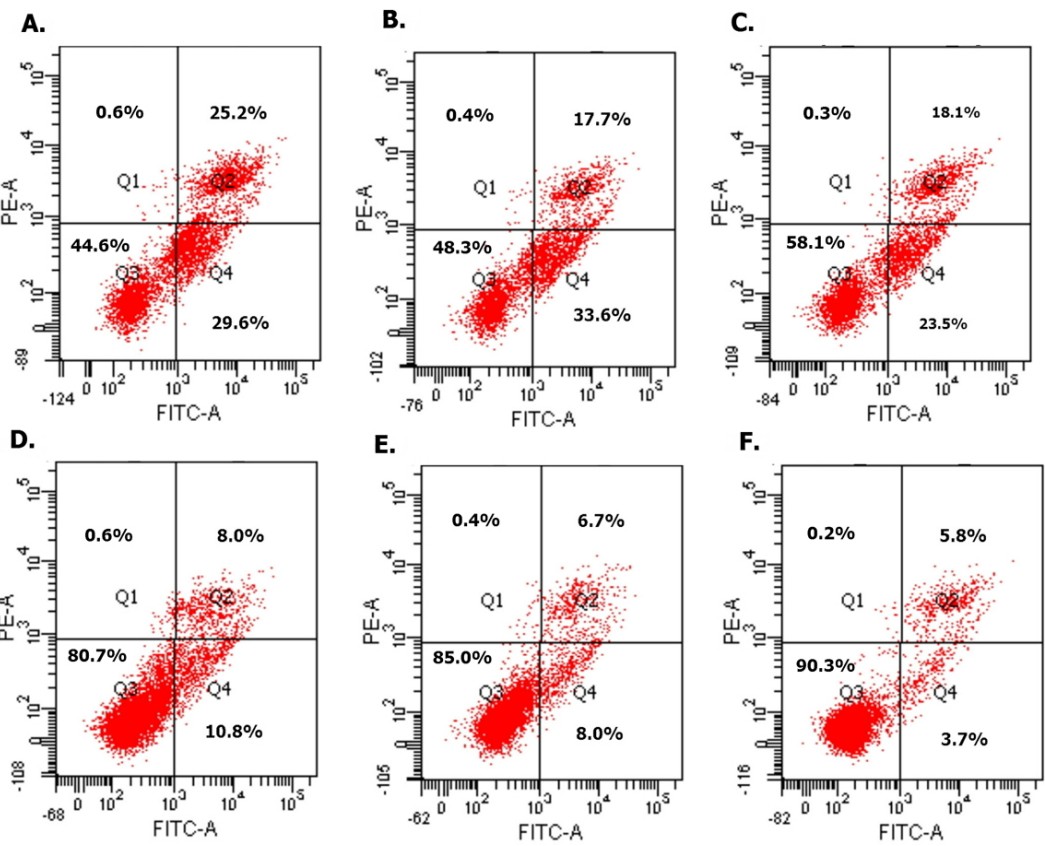

**Figure 3  Overexpressed hsa-miR-138-2-3p induced cell apoptosis after radiation by flow cytometry.** (A) and (C) show the cell apoptosis analysis of 100nM-TR and 100nMN-CR of Hep-2 cell line after radiation, respectively; (B) and (E) show the cell apoptosis analysis of 100nM-TR and 100nMN-CR of M2e cell line after radiation, respectively; (C) and (F) show the cell apoptosis analysis of 100nM-TR and 100nMN-CR of TU212 cell line after radiation, respectively. The vertical and horizontal axis stand for PI positive area and FITC positive area, respectively. Identified by flow cytometry, cells were divided into four sections: Q1: Annexin V-FITC-PI+, was representative of mechanical error; Q2: Annexin V-FITC+ PI+, was representative of late apoptosis or necrosis cells; Q3: Annexin V-FITC- PI-, was representative of living cells; Q4: Annexin V-FITC+ PI-, was representative of early apoptosis cells. (A–C) were shown the proportion of early apoptosis (29.6%, 33.6%, 23.5%) and late apoptosis (25.2%, 17.7%, 18.1%) of Hep-2, M2e, and TU212 CSCs induced by transfection of 100nM hsa-miR-138-2-3p were larger than that of induction by transfection of 100nM nonsense oligonucleotides ((D) early apoptosis of Hep-2 CSCs was 10.8%; (E) early apoptosis of M2e CSCs was 8.0%; (F) early apoptosis of TU212 CSCs was 3.7%; (D) late apoptosis of Hep-2 CSCs was 8.0%; (E) late apoptosis of M2e CSCs was 6.7%; (F) late apoptosis of TU212 CSCs was 5.8%), respectively after radiation.

of 100nM hsa-miR-138-2-3p in G1 phase (68.68%, 65.95%, 65.24%) were more than that induced by transfection of 100nM nonsense oligonucleotides (55.44%, 56.90%, 59.23%), respectively, after radiation, while the percentage of Hep-2, M2e, and TU212 CSCs induced by transfection of 100nM hsa-miR-138-2-3p in S phase (23.32%, 26.05%, 26.76%) were less than that induced by transfection of 100nM nonsense oligonucleotides (36.56%, 37.25%, 37.96%), respectively, after radiation. The percentage of M2e, and TU212 CSCs induced by transfection of 100nM hsa-miR-138-2-3p in G2 phase (8%, 8%) were more than that induced by transfection of 100nM nonsense oligonucleotides (5.85%, 2.81%),

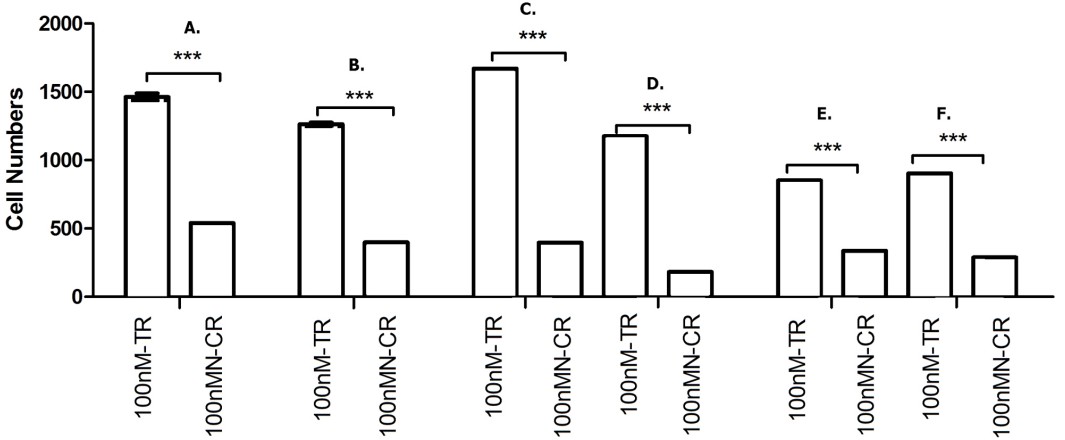

**Figure 4  Overexpressed hsa-miR-138-2-3p induced cell apoptosis after radiation.** (A) and (B) show the comparison of early and late apoptosis between 100nM-TR and 100nMN-CR of Hep-2 cell line after radiation, respectively; (C) and (D) show the comparison of early and late apoptosis between 100nM-TR and 100nMN-CR of M2e cell line after radiation, respectively; (E) and (F) show the comparison of early and late apoptosis between 100nM-TR and 100nMN-CR of TU212 cell line after radiation, respectively. The vertical and horizontal axis stand for cell numbers and early/late apoptosis, respectively. We found that the cell numbers of early apoptosis and late apoptosis of Hep-2, M2e, and TU212 CSCs induced by transfection of 100nM hsa-miR-138-2-3p were larger than that induced by transfection of 100nM nonsense oligonucleotides, respectively, after radiation, the differences between them were statistically significant (*** $P < 0.001$).

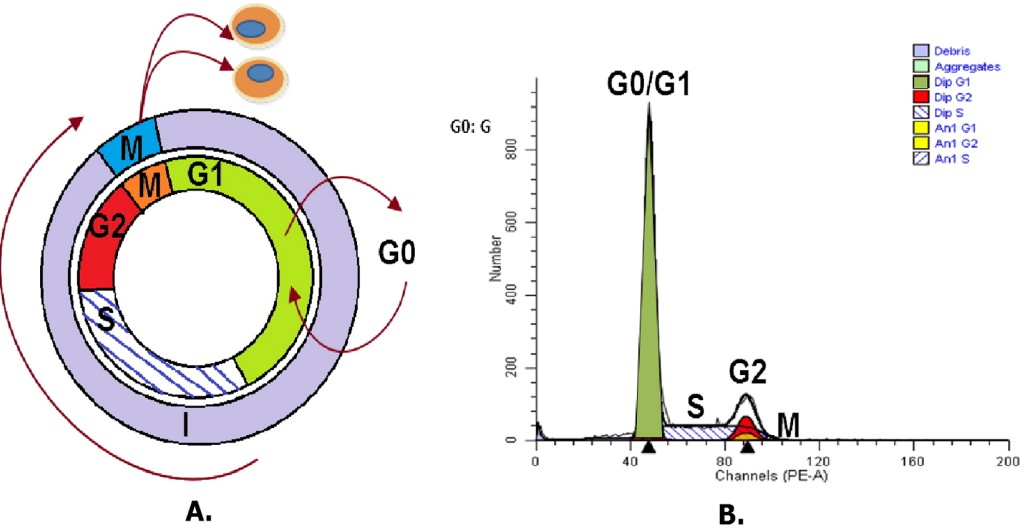

**Figure 5  Cell division cycle analysis.** As shown in (A) and (B), it is cell division cycle. G0 is the abbreviation of Gap0, cell in G0 phase has left the cycle and has stopped dividing, and cell cycle starts with this phase; G1 is the abbreviation of Gap1, cell in G1 phase increase in size, and is ready for DNA synthesis; S is the abbreviation of Synthesis, cell in S phase is for DNA replication; G2 is the abbreviation of Gap2, cell in G2 phase continue growing; M is the abbreviation of Mitosis, cell in M phase is stopped growing, and cellular energy is used for the orderly division into two daughter cells.

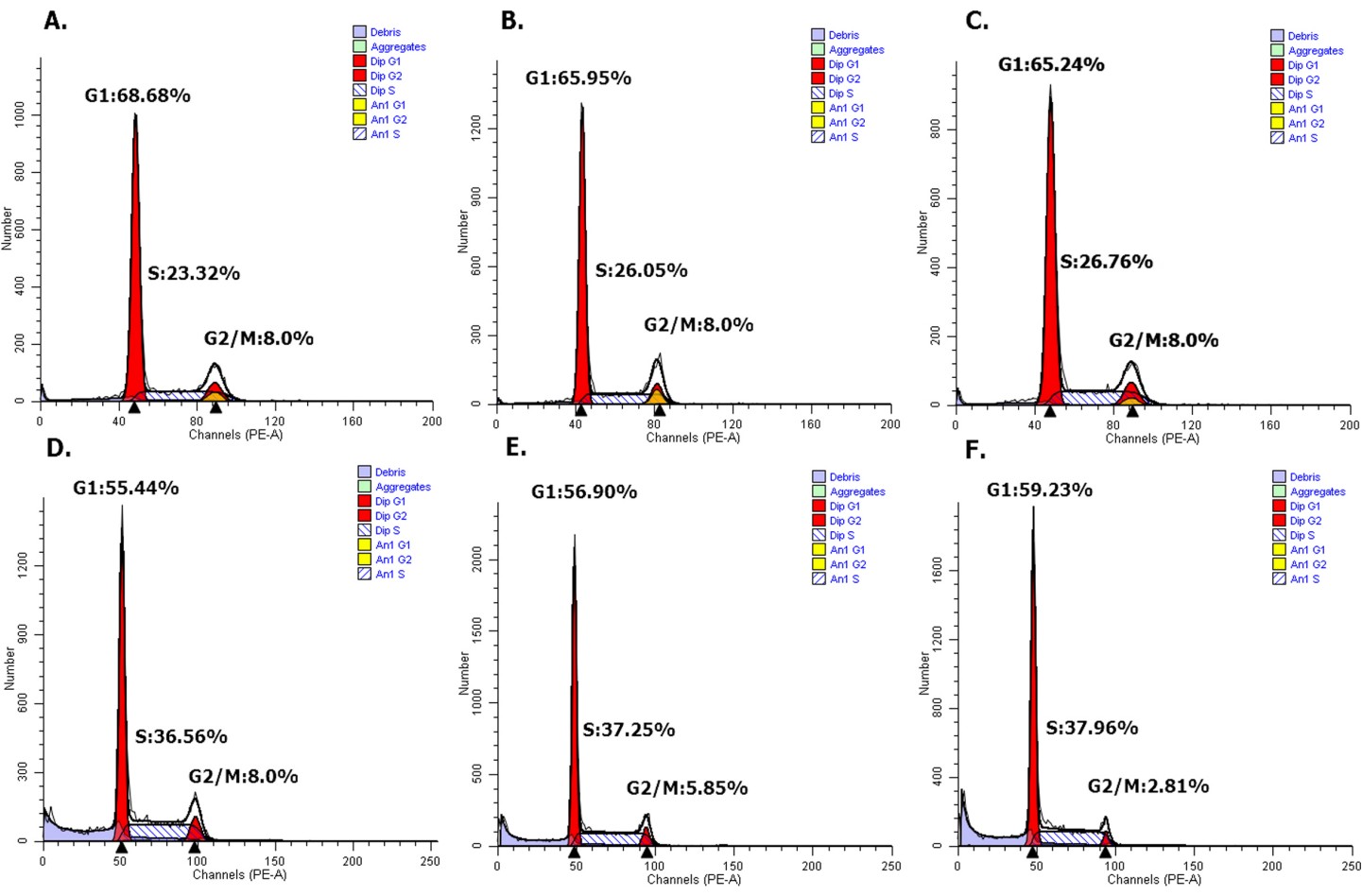

**Figure 6** **Overexpressed hsa-miR-138-2-3p arrested cell cycle at G1/S phase after radiation by flow cytometry.** (A) and (D) show the cell cycle analysis of 100nM-TR and 100nMN-CR of Hep-2 cell line after radiation, respectively; (B) and (E) show the cell cycle analysis of 100nM-TR and 100nMN-CR of M2e cell line after radiation, respectively; (C) and (F) show the cell cycle analysis of 100nM-TR and 100nMN-CR of TU212 cell line after radiation, respectively. The vertical and horizontal axis stand for cell numbers and cell division cycle, respectively. The percentage of Hep-2, M2e, and TU212 CSCs induced by transfection of 100nM hsa-miR-138-2-3p in G1 phase (68.68%, 65.95%, 65.24%) were more than that induced by transfection of 100nM nonsense oligonucleotides (55.44%, 56.90%, 59.23%), respectively after radiation. While, the percentage of Hep-2, M2e, and TU212 CSCs induced by transfection of 100nM hsa-miR-138-2-3p in S phase (23.32%, 26.05%, 26.76%) were less than that induced by transfection of 100nM nonsense oligonucleotides (36.56%, 37.25%, 37.96%), respectively, after radiation. The percentage of M2e, and TU212 CSCs induced by transfection of 100nM hsa-miR-138-2-3p in G2 phase (8%, 8%) were higher than that induced by transfection of 100nM nonsense oligonucleotides (5.85%, 2.81%), respectively, after radiation, but the percentage of Hep-2 CSCs induced by transfection of 100nM hsa-miR-138-2-3p in G2 phase (8%) was the same as that induced by transfection of 100nM nonsense oligonucleotides (8%) after radiation.

respectively, after radiation, but the percentage of Hep-2 CSCs induced by transfection of 100nM hsa-miR-138-2-3p in G2 phase (8%) was the same as that induced by transfection of 100nM nonsense oligonucleotides (8%) after radiation. However, all differences between them were not statistically significant.

### *Overexpressed hsa-miR-138-2-3p inhibited cell invasion after radiation*

Cancer migration and invasion are serious and fatal steps in cancer progression (*Zhao et al., 2012*; *Andarawewa et al., 2007*), so it is important and urgent to identify therapeutic targets to prevent the metastases of cancer cells. We found that hsa-miR-138-2-3p played a key role in inhibiting laryngeal CSCs invasion. Transwell invasion assay demonstrated that

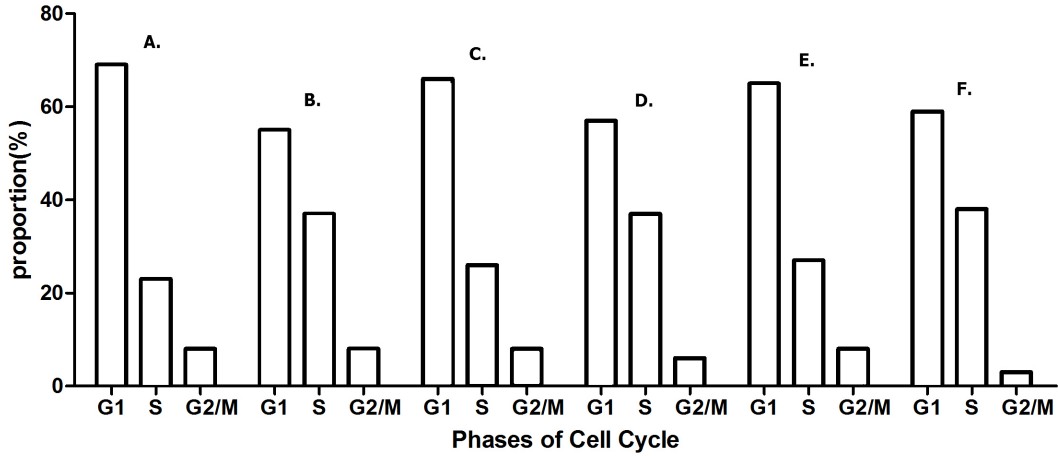

**Figure 7** **Overexpressed hsa-miR-138-2-3p arrested cell cycle at G1/S phase after radiation.** (A) and (B) show the cell cycle analysis of 100nM-TR and 100nMN-CR of Hep-2 cell line after radiation, respectively; (C) and (D) show the cell cycle analysis of 100nM-TR and 100nMN-CR of M2e cell line after radiation, respectively; (E) and (F) show the cell cycle analysis of 100nM-TR and 100nMN-CR of TU212 cell line after radiation, respectively. The vertical and horizontal axis stand for proportions and cell division phases, respectively. The proportion of Hep-2, M2e, and TU212 CSCs induced by transfection of 100nM hsa-miR-138-2-3p in G1 phase were higher than that induced by transfection of 100nM nonsense oligonucleotides, respectively after radiation. While, the percentage of Hep-2, M2e, and TU212 CSCs induced by transfection of 100nM hsa-miR-138-2-3p in S phase were less than that induced by transfection of 100nM nonsense oligonucleotides, respectively after radiation. The percentage of M2e, and TU212 CSCs induced by transfection of 100nM hsa-miR-138-2-3p in G2 phase were more than that induced by transfection of 100nM nonsense oligonucleotides, respectively after radiation, but the percentage of Hep-2 CSCs induced by transfection of 100nM hsa-miR-138-2-3p in G2 phase was the same as that induced by transfection of 100nM nonsense oligonucleotides after radiation. However, all differences between them were not statistically significant ($P > 0.05$).

the numbers of Hep-2, M2e, and TU212 CSCs that were transfected into 50nM, 100nM, and 150nM hsa-miR-138-2-3p penetrated through matrigel were lower than that were transfected into 100nM nonsense oligonucleotides and PBS buffer at 24 h after radiation, a significant difference between experimental teams and control teams (Figs. 8 and 9). It was important to note that the cell numbers of 100nM-TR and 150nM-TR were lower than 50nM-TR and the differences were statistically significant. However, this phenomenon was not investigated without radiation, the cell numbers of 100nM-C were not lower than 100nMN-C and PBS-C (Figs. 8 and 9).

***Overexpressed hsa-miR-138-2-3p reduced survival fraction after radiation***
100nM-TR, 100nM-CR, and non-transfection of Hep-2 cell line were treated with 0, 2, 4, 6, 8 Gy X-ray irradiation, respectively, and then were incubated for additional 14 days. After incubation, cells were fixed with paraformaldehyde and the colonies (>50 cells) were counted for each culture dish by a light microscope. The survival fraction (SF) of 100nM-TR, 100nM-CR, and non-transfection of Hep-2 CSCs were evaluated. The SF of 100nM-TR of all laryngeal CSCs were lower than that of 100nMN-CR and non-transfection, and the difference between 100nM-TR and non-transfection were statistically significant (Fig. 10).

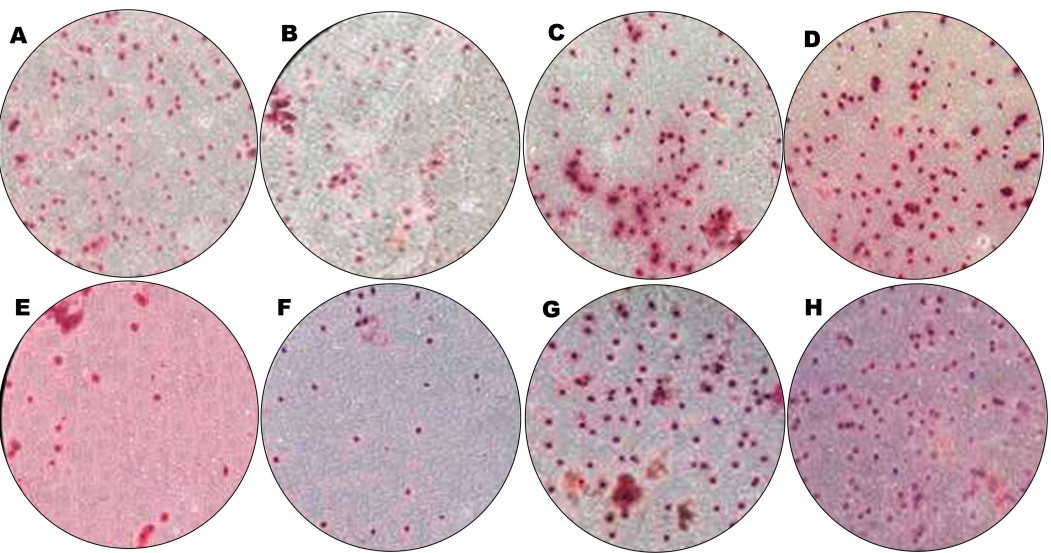

**Figure 8** **Overexpressed hsa-miR-138-2-3p inhibited cell invasion by Transwell assay.** (A–C) show Transwell Assay of 100nM-T, 100nMN-C and PBS-C of Hep-2 cell line without radiation, respectively; (D–H) show Transwell Assay of 50nM-TR, 100nM-TR, 150nM-TR, 100nMN-CR and PBS-CR of Hep-2 cell line 24 h after radiation, respectively. The cell numbers of experimental teams and control teams of Hep-2 CSCs were reduced at 24 h after radiation. Among them, Hep-2 CSCs were dropped most. The cell number of 50nM-TR, 100nM-TR, and 150nM-TR of Hep-2 CSCs were (121.6 ± 4.62), (41.6 ± 4.62) and (40.8 ± 3.63) were lower than 100nMN-CR (168.80 ± 23.2) and PBS-CR (187.08 ± 24.6) of that, respectively. It was important to note that the cell numbers of 100nM-TR and 150nM-TR were lower than 50nM-TR and the differences were statistically significant. However, this phenomenon was not found in non-radiation, the cell numbers of 100nM-T, 100nMN-C, and PBS-C were (212.6 ± 29.9), (226.6 ± 32.2), and (228.2 ± 33.5), respectively.

***Overexpressed hsa-miR-138-2-3p promoted DNA damage after radiation***

The degree of DNA damage after radiation was measured by comet assay. As shown in Fig. 11, the appearance of "comet" with fragmented DNA (tail) being separated from undamaged nuclear DNA (head) was seen in 100nM-TR and 100nMN-CR of Hep-2, M2e, and TU212 CSCs after radiation. We found that the "heads" of "comet" of 100nM-TR were smaller than that of 100nMN-CR, while the "tails" of "comet" of 100nM-TR were longer than that of 100nMN-CR. Further, the measurements were made by Comet Assay IV software to calculate the tail movement (%). As Fig. 12 shown, the tail movement of 100nM-TR of all laryngeal CSCs were higher than that of 100nMN-CR, the differences between 100nM-TR and 100nMN-CR of Hep-2 cells were statistically significant. These data were indicated that the DNA damage of 100nM-TR were more serious than that of 100nM-CR in laryngeal CSCs after radiation.

## Overexpressed hsa-miR-138-2-3p regulated signal transduction pathway of laryngeal CSCs after radiation
### Overexpressed hsa-miR-138-2-3p inhibited Wnt/β-catenin pathway

It was known that accumulation of β-catenin leaded to abnormal activation of Wnt/β-catenin signaling pathway and reinforce radiation resistance. As shown in Figs. 13 and 14, the expression of β-catenin in 100nM-TR of all laryngeal CSCs were reduced more than that

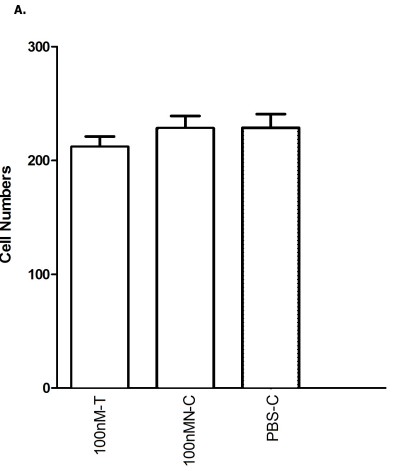

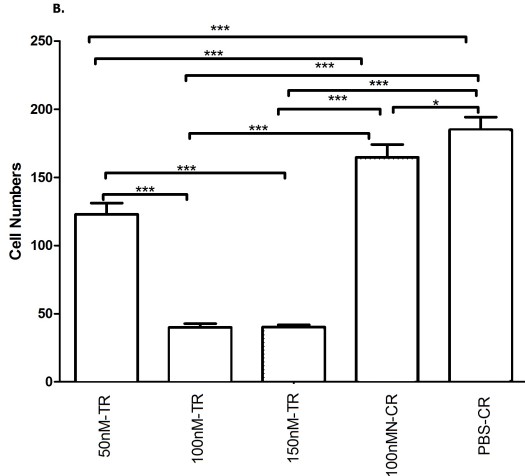

**Figure 9** **Overexpressed hsa-miR-138-2-3p inhibited laryngeal CSCs invasion.** (A) shows the comparison of Transwell Assay among 100nM-T, 100nMN-C and PBS-C of Hep-2 cell line without radiation, respectively; (B) shows the comparison of Transwell Assay of 50nM-TR, 100nM-TR, 150nM-TR, 100nMN-CR and PBS-CR of Hep-2 cell line 24 h after radiation, respectively. The vertical and horizontal axis in the (A) and (B) stand for cell numbers and treatments, respectively. Transwell invasion assay demonstrated that the numbers of Hep-2 CSCs that were transfected into 50nM, 100nM, and 150nM hsa-miR-138-2-3p penetrated through matrigel were lower than that were transfected into 100nM nonsense oligonucleotides and PBS buffer at 24 h after radiation, a significant difference between experimental teams and control teams (*** $P < 0.001$) (B). It was important to note that the cell numbers of 100nM-TR and 150nM-TR were lower than 50nM-TR and the differences were statistically significant (*** $P < 0.001$). However, this phenomenon was not investigated without radiation, the cell numbers of 100nM-C were not significantly lower than 100nMN-C and PBS-C ($P > 0.05$) (Fig. 9A).

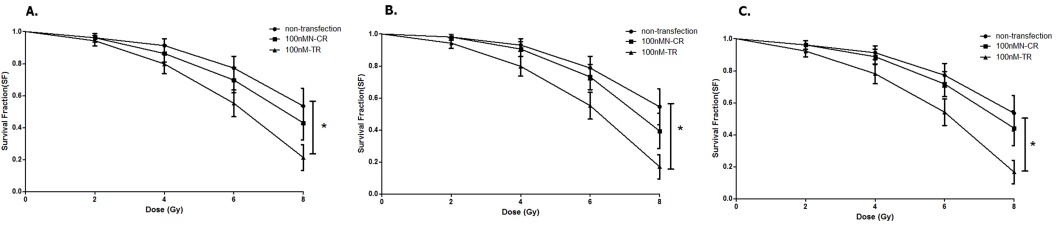

**Figure 10** **Overexpressed hsa-miR-138-2-3p reduced survival fraction after radiation.** (A–C) shows the comparison of survival analysis among 100nM-TR, 100nMN-CR and non-transfection of Hep-2, M2e, and TU212 cell lines after radiation, respectively. The vertical and horizontal axis stand for survival fraction and does(Gy), respectively. 100nM-TR, 100nM-CR, and non-transfection of Hep-2, M2e and TU212 were treated with 0, 2, 4, 6, 8 Gy X-ray irradiation, respectively, and the survival fraction of 100nM-TR of all laryngeal CSCs were lower than that of 100nMN-CR and non-transfection, and the difference between 100nM-TR and non-transfection were statistically significant (* $P < 0.05$).

in 100nMN-CR, and the results between 100nM-TR and 100nMN-CR of all laryngeal CSCs were significant difference. It suggested that overexpression of hsa-miR-138-2-3p reduced the expression of $\beta$-catenin, further inhibited the activity of Wnt/$\beta$-catenin pathway and reduced the resistance of the laryngeal CSCs to radiation.

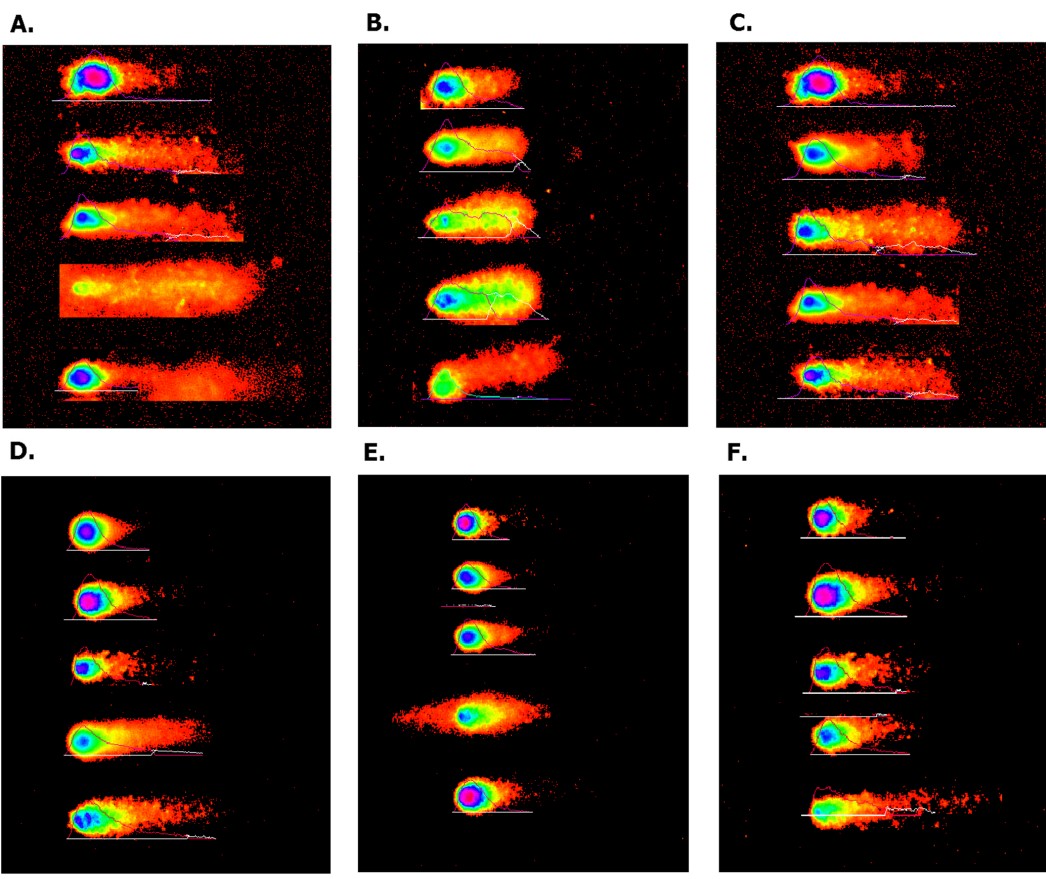

**Figure 11** **Overexpressed hsa-miR-138-2-3p promoted DNA damage after radiation by Comet assay.**
(A) and (D) show the DNA damage analysis of 100nM-TR and 100nMN-CR of Hep-2 cell line after radia-
tion, respectively; (B) and (E) show the DNA damage analysis of 100nM-TR and 100nMN-CR of M2e cell
line after radiation, respectively; (C) and (F) show the DNA damage analysis of 100nM-TR and 100nMN-
CR of TU212 cell line after radiation, respectively. The appearance of "comet" with fragmented DNA
(tail) being separated from undamaged nuclear DNA (head) was seen in 100nM-TR and 100nMN-CR of
Hep-2, M2e, and TU212 CSCs after radiation. It was found that the "heads" of "comet" of 100nM-TR
were smaller than that of 100nMN-CR, while the "tails" of "comet" of 100nM-TR were longer than that
of 100nMN-CR. These data were indicated that the DNA damage of 100nM-TR were more serious than
that of 100nM-CR in laryngeal CSCs after radiation.

### Overexpressed hsa-miR-138-2-3p inhibited Hippo/YAP1 pathway

Hippo signal pathway regulated cell prolifertion, development and progression. Ancillary
transcription factor YAP1 played a key role in Hippo pathway. Knocking out nuclear YAP1
can inhibit tumor cell proliferation, while increased expression of YAP1 can promote cell
growth and migration and inhibit apoptosis. As shown in Figs. 15 and 16, the expression
of YAP1 in 100nM-TR of all laryngeal CSCs were down-regulated much more than that
in 100nMN-CR, and the results between 100nM-TR and 100nMN-CR of all laryngeal
CSCs were significant difference. It suggested that overexpression of hsa-miR-138-2-3p
reduced the expression of YAP1, further controlling Hippo signal pathway to weaken
radio-resistance in laryngeal CSCs.
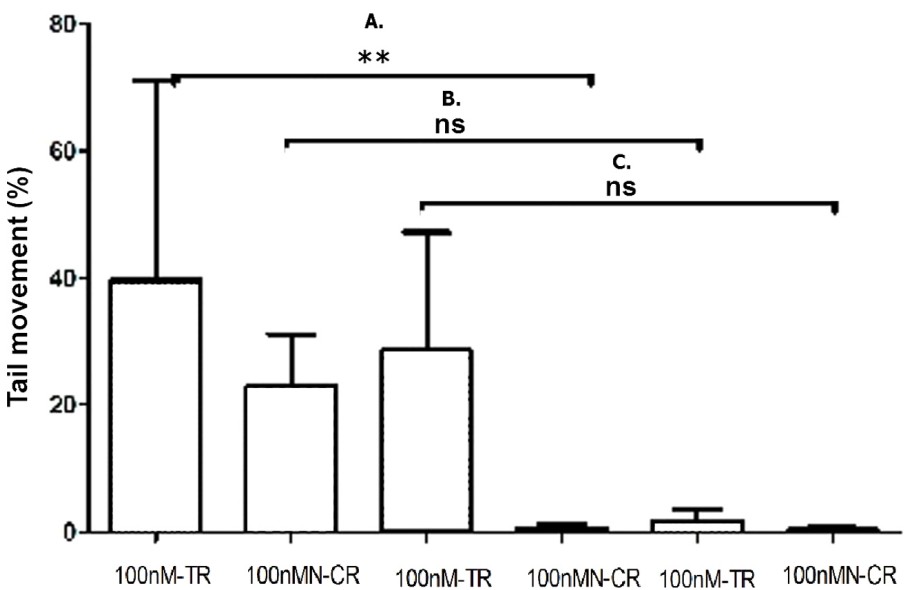

**Figure 12 Overexpressed hsa-miR-138-2-3p promoted DNA damage after radiation.** (A) shows the comparison of DNA damage between 100nM-TR and 100nMN-CR of Hep-2 cell line after radiation, respectively; (B) shows the comparison of DNA damage between 100nM-TR and 100nMN-CR of M2e cell line after radiation, respectively; (C) shows the comparison of DNA damage between 100nM-TR and 100nMN-CR of TU212 cell line after radiation, respectively. The vertical and horizontal axis stand for Tail movement (%) and treaments, respectively. The tail movement of 100nM-TR of all laryngeal cell lines were higher than that of 100nMN-CR, and the differences between 100nM-TR and 100nMN-CR of Hep-2 CSCs were statistically significant (** $P < 0.01$). These data were indicated that the DNA damage of 100nM-TR were more serious than that of 100nM-CR in laryngeal CSCs after radiation.

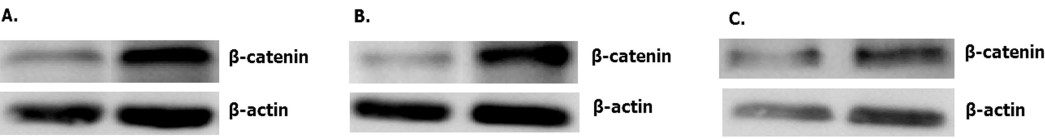

**Figure 13 Overexpressed hsa-miR-138-2-3p down-regulated expression of $\beta$-catenin.** The expression of $\beta$-catenin in 100nM-TR of Hep-2 (A), M2e (B) and TU212 (C) CSCs were reduced more than that in 100nMN-CR.

### Overexpressed hsa-miR-138-2-3p activated JNK1/p38/MAPK pathway

MAPK/JNK signal pathway was activated by radiation, and enhanced the radio-sensitivity of tumor cells by its association with radiation-induced DNA damage (*Lagadec et al., 2012*; *Trairut & Slack, 2013*). As shown in Figs. 17–19, the expression of JNK1 and p38 in 100nM-TR of all laryngeal CSCs were improved much more than that in 100nMN-CR, and the results between 100nM-TR and 100nMN-CR of all laryngeal CSCs were significant difference. It suggested that overexpression of hsa-miR-138-2-3p promoted the expression of JNK1 and p38, further activating JNK1/p38/MAPK signal pathway to increase radio-sensitivity in laryngeal CSCs.

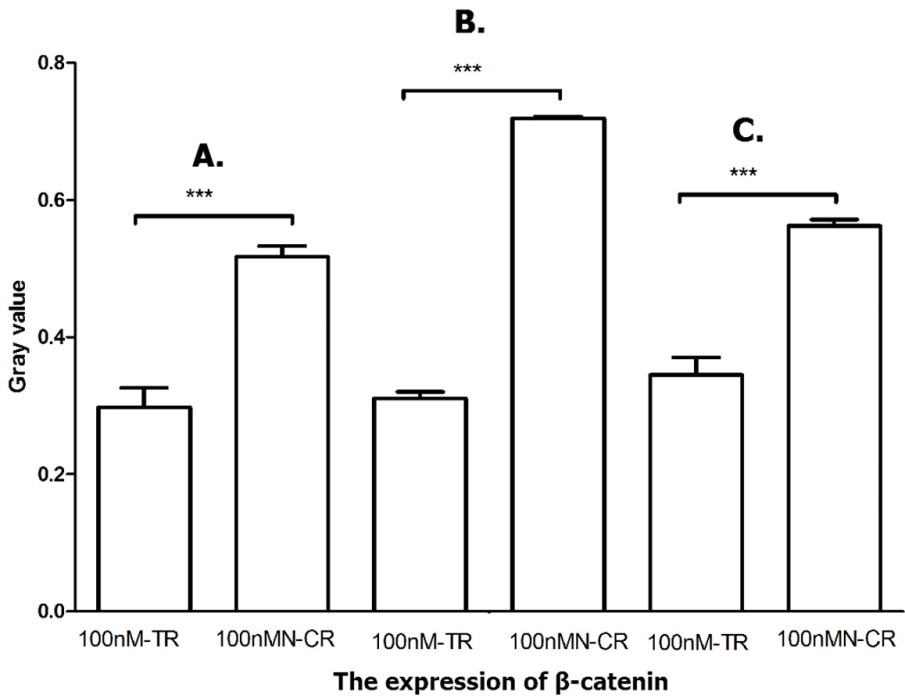

**Figure 14** **Overexpressed hsa-miR-138-2-3p inhibited Wnt/β-catenin pathway.** The vertical and horizontal axis stand for Gary value and treaments, respectively. It suggested that the expression of β-catenin between 100nM-TR and 100nMN-CR of Hep-2 (A), M2e (B) and TU212 (C) CSCs were significant difference (*** $P < 0.001$).

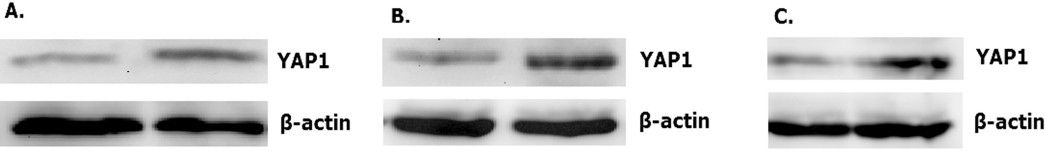

**Figure 15** **Overexpressed hsa-miR-138-2-3p down-regulated expression of YAP1.** The expression of YAP1 in 100nM-TR of Hep-2 (A), M2e (B) and TU212 (C) CSCs were reduced more than that in 100nMN-CR.

## DISCUSSION

In recent years, a large number of studies have focused on miRNA and cancer radiosensitivity. miRNAs played important roles in cell proliferation, invasion, apoptosis and cell cycle arrest, and aggressive growth, development, and metastasis are another characteristics of radio-resistance (*Lagadec et al., 2012*; *Andarawewa et al., 2007*). *Balça-Silva et al. (2012)* found that over-expressed miR-34 enhanced the radio-sensitivity of non-small cell lung cancers. In gastric cancer, knocking out miR-221 and miR-222 could inhibit proliferation and invasion, increase radiosensitivity of gastric carcinoma cells (*Zhang et al., 2010*). So far, however, there is no report on the relationship between hsa-miR-138-2-3p and radiation sensitivity of laryngeal cancer. In our present study, overexpressed hsa-miR-138-2-3p played an key role in decreasing laryngeal CSCs proliferation and invasion;

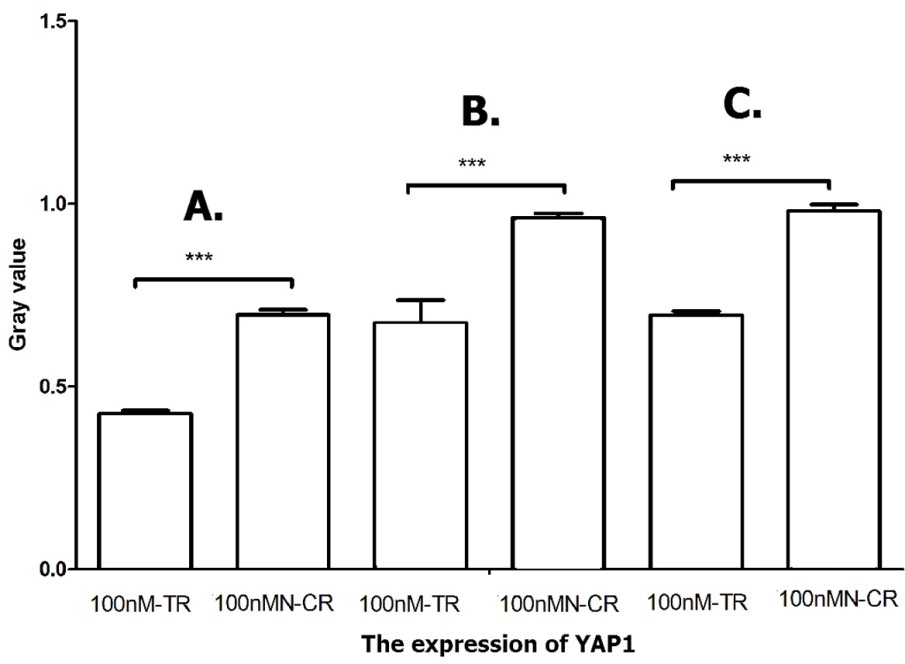

**Figure 16  Overexpressed hsa-miR-138-2-3p inhibited Hippo/YAP1 pathway.** The vertical and horizontal axis stand for Gary value and treaments, respectively. It suggested that the expression of YAP1 between 100nM-TR and 100nMN-CR of Hep-2 (A), M2e (B) and TU212 (C) CSCs were significant difference (*** $P < 0.001$).

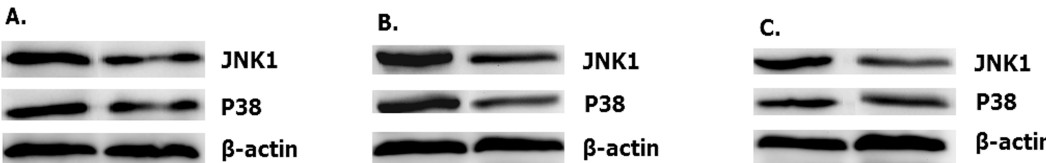

**Figure 17  Overexpressed hsa-miR-138-2-3p up-regulated expression of JNK1 and p38.** The expression of JNK1 and p38 in 100nM-TR of Hep-2 (A), M2e (B) and TU212 (C) CSCs were improved more than that in 100nMN-CR.

increasing the proportion of early and late apoptosis in laryngeal CSCs; raising G1 phase arrest; and down-regulating the proportion of S stage cells of cell cycle that were related to radio-resistance in laryngeal CSCs.

Wnt/$\beta$-catenin signal transduction pathway regulated many cellular processes such as cell proliferation, apoptosis and aggressiveness, of which, $\beta$-catenin is the major factor. Che et al. (*Andarawewa et al., 2007*) investigated that Cox-2 inhibitor NS398 inhibited the expression of DNA-PKCs and controlled Wnt/ $\beta$-catenin pathway to improve the radiation sensitivity of Eca109 cells. *Chang et al. (2016)* suggested that the radiation resistance of AMC-HN-9 cells were decreased remarkably when $\beta$-catenin was knocked out. Wang's study (*Wang et al., 2014a*) showed that the excessive expression of lncRNA–p21 improve the radiosensitivity of cells by inhibiting Wnt/ $\beta$-catenin signaling pathway. Regulating upstream signaling molecules of Wnt/$\beta$-catenin pathway and promoting degradiation of $\beta$-catenin may become new anticancer treatments (*Huang et al., 2012a*; *Qi et al., 2012*;

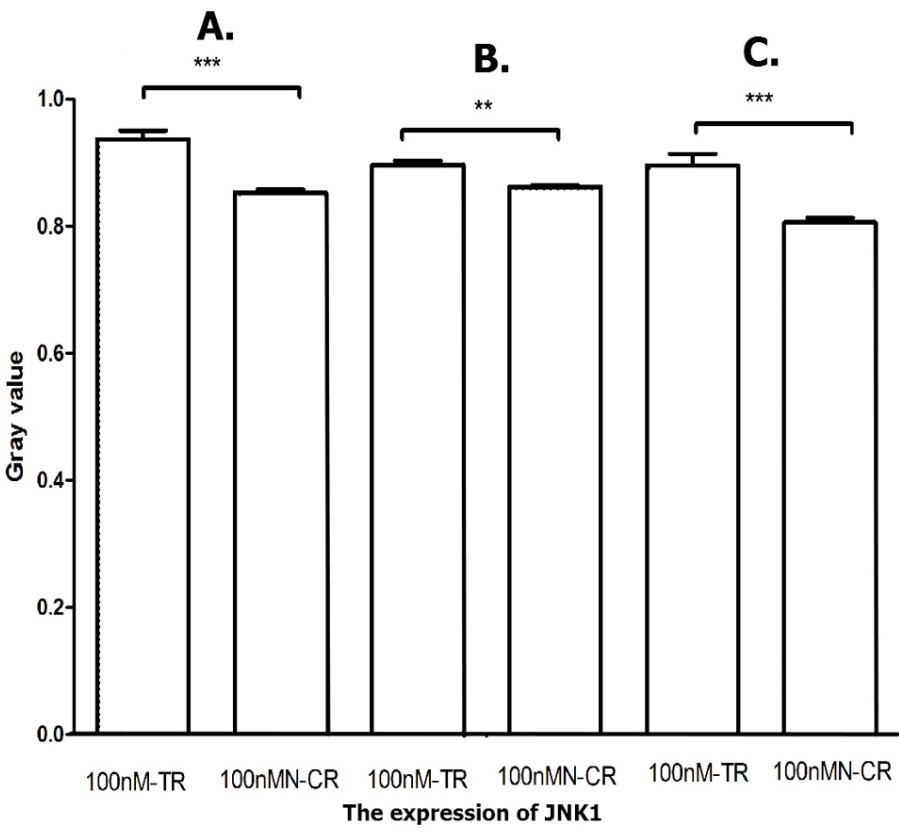

**Figure 18** **Overexpressed hsa-miR-138-2-3p activated JNK1/MAPK pathway.** The vertical and horizontal axis stand for Gary value and treaments, respectively. It suggested that the expression of JNK1 between 100nM-TR and 100nMN-CR of Hep-2 (A), M2e (B) and TU212 (C) CSCs were significant difference (** $P < 0.01$, *** $P < 0.001$).

*Wang et al., 2013*). Recent studies have revealed that Hippo signaling pathway played a role in the growth of many types of cancer cells, and YAP1 was associated with lung cancer and ovarian malignant tumors. Cell proliferation was inhibited by knocking out YAP1, while increased expression of YAP1 accelerated cell growth and migration. YAP1 was also called oncoprotein. The MAPK signal pathway plays a role in regulating many cellular activities, such as growth, differentiation and stress reaction. Many signal transduction pathways related to radiation were regulated by the MAPK family, including extracellular signal regulating kinase (ERK), c-Jun amino end kinase (JNK) and the p38 MAPK pathway. JNK1 and JNK2, the stress inducing protein kinases, induce the phosphorylation of transcription factor c-Jun. A MAPK pathway activated in this way is also called stress activating protein kinase (SAPK) pathway. Studies have shown that apoptosis initiation and cell cycle arrest were closely related to the MAPK/p38 and MAPK/JNK1 pathway. Mediating the MAPK/p38 and MAPK/JNK1 pathways can affect the radio-sensitivity of tumor cells. The MAPK signaling pathway is considered to have an important effect on the radiation sensitivity of tumor cells for its association with radiation-induced DNA damage (*Lagadec et al., 2012*). The activation of JNK and associated signaling pathway is related to apoptosis. Activation of SAPK/JNK leads to cell radio-sensitivity, and vice versa

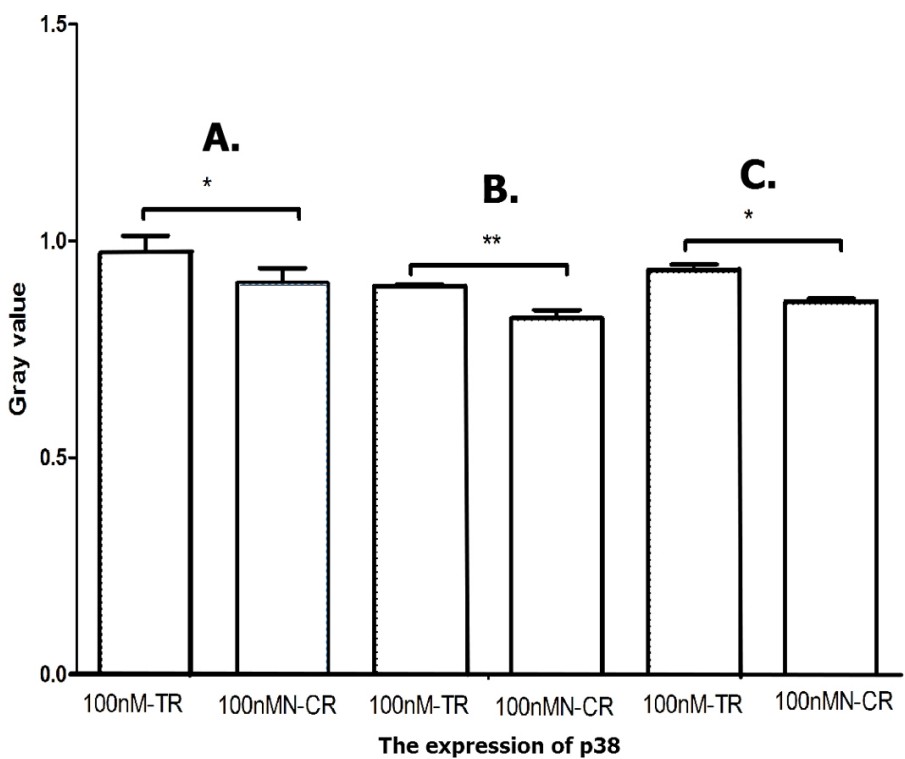

**Figure 19  Overexpressed hsa-miR-138-2-3p activated p38/MAPK pathway.** The vertical and horizontal axis stand for Gary value and treaments, respectively. It suggests that the expression of p38 between 100nM-TR and 100nMN-CR of Hep-2 (A), M2e (B) and TU212 (C) CSCs were significantly difference (* $P < 0.05$, ** $P < 0.01$).

(*Trairut & Slack, 2013*). *Wang et al. (2013)* and Bulavin et al. (*Kobayashi et al., 2014a*) found that MAPK/p38 signaling pathway regulated the transition of G2 / M phase in mammalian cells and participated in G2 arrest. *Wakita et al. (2015)* showed that Reg Iα-expressing cells activated the MAPK/JNK1 pathway to increase JNK1 protein expression so as to improve the radio-sensitivity of esophageal squamous cell carcinomas. The present study showed that overexpression of hsa-miR-138-2-3p reduced the expressions of $\beta$-catenin and YAP1 in the laryngeal CSCs after radiation, further inhibited Wnt/ $\beta$-catenin and Hippo/YAP1 signal pathways to weaken radio-resistance in laryngeal CSCs.

## CONCLUSION

The present research indicated that overexpressed hsa-miR-138-2-3p play key role in decreasing laryngeal CSCs proliferation and invasion; increasing the proportion of early and late apoptosis in laryngeal CSCs; raising G1 phase arrest; and down-regulating the proportion of S stage cells of cell cycle that were related to radio-resistance in laryngeal CSCs. Overexpressed hsa-miR-138-2-3p regulated signal transduction pathway of laryngeal CSCs after radiation. Over-expression of hsa-miR-138-2-3p down-regulated the expression of $\beta$-catenin and YAP1 in the laryngeal CSCs after radiation, further inhibited Wnt/$\beta$-catenin and Hippo/YAP1 signal pathways to weaken radio-resistance in laryngeal CSCs. While the

expressions of JNK1 and p38 were promoted, the JNK1/p38/MAPK signal pathway was activated to increase radio-sensitivity in laryngeal CSCs. These results are useful for a better understanding of hsa-miR-138-2-3p in laryngeal CSCs, and prove hsa-miR-138-2-3p as a promising biomarker and as a target for diagnosis and for novel anti-cancer therapies for laryngeal cancers.

### Funding

The study was supported by grants from the Nature Science Foundation of China (#81072495), the Science and Technology Key Projects of Zhejiang province, China (#2010C33006; #2017C03053), the Science and Technology Projects of Hangzhou, China (#20140733Q18, #20150733Q22). The funders had no role in study design, data collection and analysis, decision to publish, or preparation of the manuscript.

### Grant Disclosures

The following grant information was disclosed by the authors:
Nature Science Foundation of China: #81072495.
Science and Technology Key Projects of Zhejiang province, China: #2010C33006, #2017C03053.
Science and Technology Projects of Hangzhou, China: #20140733Q18, #20150733Q22.

### Competing Interests

The authors declare there are no competing interests.

### Author Contributions

- Ying Zhu and Li-Yun Shi conceived and designed the experiments, performed the experiments, wrote the paper.
- Yan-Min Lei and Yan-Hong Bao performed the experiments, analyzed the data.
- Zhao-Yang Li and Fei Ding analyzed the data, contributed reagents/materials/analysis tools, reviewed drafts of the paper.
- Gui-Ting Zhu contributed reagents/materials/analysis tools, prepared figures and/or tables, reviewed drafts of the paper.
- Qing-Qing Wang prepared figures and/or tables, reviewed drafts of the paper.
- Chang-Xin Huang conceived and designed the experiments, wrote the paper.

### Data Availability

The raw data has been supplied as Data S1.

### Supplemental Information

Supplemental information for this article can be found online at http://dx.doi.org/10.7717/peerj.3233#supplemental-information.

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
