# Peer review of "Radiosensitization effect of hsa-miR-138-2-3p on human laryngeal cancer stem cells"

_PeerJ, doi:10.7717/peerj.3233_

## Round 0.1 · original submission · Minor Revisions

The author presented good work on the miR-138, and it will be improved by addressing the comments from reviewers. Also, you should have the English language edited by a native speaker.

Reviewer 1 ·

Basic reporting

The manuscript discusses radiosensitization effect of miR-138 on laryngeal cancer stem cells. As previous studies showed, miR-138 may have an important role in the self-regulation of laryngeal squamous cancer stem cell radiation sensitivity. This study further investigate its effect by a set of well-defined experiments. Overall, the paper is well-written and well-organized. Below are my comments:

1. In this manuscript, there are two different terms “miR-138” and “miR-138-2”. The authors should make it clear what’s the relationship/difference between them (if there is any).

2. General comments on the figures: please double check all figures to make sure that 1) it has enough high resolution; 2) the caption is clear enough to explain the figure.

Experimental design

Overall, the experiments are well-defined, supported with solid analysis. Below are places that should be improved:

1. It’s not very clear how transient transfection is done. The authors should extend the sentence (lines 132-135) and give more details.

2. In line 156, the sentence “after various times” is not clear. Please clarify.

Validity of the findings

The analyses are solid and well supported with data. I have no comments on this part.

Additional comments

Below are general specific comments:

The manuscript summarized 5 main results in abstract (line 39-51). To improve the readability, I suggest to organize the subsection with numbers like (1), (2),..., instead of using line breaks.

Introduction is organized as a single long paragraph. I suggest to split this section into multiple paragraphs appropriately to improve readability.

Line 66: It is not common to cite an article with both last name and first name of an author. Suggested change: Rycaj et al, instead of Rycaj K et al.

Line 80: Citation needed for the sentence “MicroRNA (miRNA), a non-coding ...”.

Line 82 - 87: the manuscript mentioned that “In our previous research, …...”. But there is no citation/reference related to the sentences.

Line 109: Define “CSC” where the full name appears for the first time.

Line 228: It is not clear to me what does the part included in the parentheses. It seems to be some wrong-place text.

Line 246: The term in vitro should be italicized as in vitro. Also true for line 252.

Figure 1: The third sub figure is in bad quality. Same comment applies to Figure 4. Suggestion: remake the figure and use large font size to make sure that all figures have enough resolution.

Figure 2: It’s better to place all the sub figures into a single figure so that they are not separated into multiple pages when editing. Also, please explain what is A450 either in the main text or in the caption.

Line 313: missing citation for ModFit software.

Figure 9: In Figure 2, the labels for sub figures (i.e., A, B, C, D) are placed on the upper left corner of each sub figure. Figure 9 should follow the same format.

Line 408: change “figure” into “Figure”.

Line 409 & 434: remove extra comma.

Line 421: In the sentence “As shown in Figure 14 hrs after ...”, it seems that something is missing before hrs.

In the text, some places use “hr” (e.g., line 135) while other places use “h” (e.g. 272), please be consistent.

Reference sections: please use consistent format for all references. In the current version of the manuscript, the two major inconsistencies are: 1) Author names - please use author full name (e.g., Claudia Peizsch) or abbreviated form for first name (e.g., Rycaj K), do not mix them; 2) Most of the journal names have abbreviated form, a few of them have full name, please double check if these journals have appropriate abbreviated form or not.

·

Basic reporting

The article meets the standards of the journal, it is clear clear and well structured.

Experimental design

no comment

Validity of the findings

no comment

Additional comments

I want the authors to rewrite the conclusion for the article.

Reviewer 3 ·

Basic reporting

Overall, the manuscript by Zhu et.al., described their experiments and results. logically and clearly.
The study was well designed and had appropriate methodology. While the need work more their professional English as a scientific article. Please see me detail comments below.

Experimental design

No comment

Validity of the findings

No comment

Additional comments

1. Is there any previous studies on hsa-miR-138? The author should mention the preview cancer studies on hsa-miR138-2-3p to give more background on why they focus on it.
2. In the manuscript, the authors did over-expression with miR-138-2-3p and tested the phenotypes based on the transfections. While, in their manuscript, the authors presented and discussed the results using “miR-138” , “miR-138-2” , “miR-138-2-3p” and “miR-138-2 mimics” randomly. According to miRBase, the slightly difference on their names actually represent different sequence , which may have different biology functions.
3.Please check and clarify the right name to used in the whole manuscript. In additional, the author should specify human miR-138-2-3p, which they did experiments on, using the right name as “hsa-miR138-2-3p”.
4. The English language should be improved to ensure the manuscript described their works in a scientific manner. For example, in their Abstract (Page1 ), all the results or sentence listed in the results section, are not sounds solid with “might” ,” could ” and so on. Please specific the observation and conclusion from your experiments and write them in a confidence way and then put the interpretation in discussion. Except Figure2, all the other figures lack figure legends to explain what the figure means and what the axises represent for.

---

## Round 0.2 · Minor Revisions

This version is almost ready to be Accepted. However, before final Acceptance please modify the manuscript according to the reviewer's final comments, which are mostly copyedits.

Reviewer 1 ·

Basic reporting

No comment

Experimental design

No comment.

Validity of the findings

No comment.

Additional comments

All my previous comments have been properly addressed by the authors in the revised version.

Reviewer 4 ·

Basic reporting

The authors have made a careful revision according to the reviewers' comments. But the authors should further improve the English expression.

Experimental design

no comment.

Validity of the findings

no comment.

Additional comments

My detailed comments are as below:
1. In Materials and Methods section, some places use “Hep-2, TU212 and M2e” (e.g., line 94) while other places use “HEP-2, M2E, and TU212”. Please stay consistent.
2. Some places use “were” (e.g., line 176) while other places use “was” (e.g., line 177). Please stay consistent.
3. Please rewrite the sentence “we appied starBase v2.0 (http://starbase.sysu.edu.cn/index.php)[13], and signal pathways that were controlled by miRNAs were referred from 13 pathway databases (including GO, KEGG, and BIOCARTA).”
4. In the text, there is still “hr”.
5. Please rewrite the sentence “The sequence of these oligonucleotides were shown in Table 2.”
6. Please rewrite the sentence “As Figure 2 shown, the cell proliferation of 50nM-TR, 100nM-TR, 150nM-TR, 100nMN-CR, and PBS-CR of HEP-2 cell lines at 0h, 24h, 48h and 72h after radiation.”
7. Please delete “Date are reported as mean±SD.” in line 264.
8. The sentence “While, the expression of JNK1 and p38 were promoted, and further activating JNK1/p38/MAPK signal pathway to increase radio-sensitivity of laryngeal CSCs.” in line 567-569 has been repeated in Conclusion.
9. In reference section, there are “[J]” after the titles.

---

## Round 0.3 · accepted · Accept

This is a great work about the microRNAs on human laryngeal cancer stem cells.